# Use of Diatoms in Monitoring the Sakarya River Basin, Turkey

**Cüneyt Nadir Solak** [1],*, **Łukasz Peszek** [2], **Elif Yilmaz** [1], **Halim Aytekin Ergül** [3], **Melih Kayal** [4], **Fatih Ekmekçi** [4], **Gábor Várbíró** [5], **Arzu Morkoyunlu Yüce** [6], **Oltan Canli** [7], **Mithat Sinan Binici** [7] **and Éva Ács** [5,8]

1. Department of Biology, Arts and Science Faculty, Dumlupınar University, 43100 Kütahya, Turkey; elfyilmaz38@gmail.com
2. Department of Agroecology, Institute of Agricultural Sciences, Land Management and Environmental Protection, University of Rzeszów, Zelwerowicza 8B, 35–601 Rzeszów, Poland; lukaspeszek@gmail.com
3. Department of Biology, Science and Literature Faculty, Kocaeli University, 41380 Kocaeli, Turkey; halim.ergul@gmail.com
4. Environmental Section, Investigating, Planning and Allocations Department, General Directorate of State Hydraulic Works, 06100 Ankara, Turkey; melihkayal@dsi.gov.tr (M.K.); fekmekci@dsi.gov.tr (F.E.)
5. MTA Centre for Ecological Research, Danube Research Institute, Karolina út 29, H-1113 Budapest, Hungary; varbiro.gabor@okologia.mta.hu (G.V.); acs.eva@okologia.mta.hu (E.A.)
6. Hereke O. İ. Uzunyol Vocational School, Kocaeli University, 41800 Kocaeli, Turkey; arzuyuce38@gmail.com
7. TUBITAK Marmara Reserach Centre, Environment and Cleaner Production Institute, 41470 Kocaeli, Turkey; oltan.canli@tubitak.gov.tr (O.C.); sinan.binici@tubitak.gov.tr (M.S.B.)
8. Faculty of Water Sciences, National University of Public Service, Bajcsy-Zsilinszky utca 12-14, H-6500 Baja, Hungary
* Correspondence: cnsolak@gmail.com

**Abstract:** The Sakarya River basin is one of the largest basins in Turkey, and encompasses the Kocaeli, Düzce, Sakarya, Bursa, Bilecik, Bolu, Kütahya, Eskişehir, Ankara, Afyon, and Konya provinces. In this study, the water quality status of the basin was investigated using 18 diatom indices, calculated in Omnidia software. For this purpose, a total of 46 stations were surveyed in the rivers and streams of the basin in May 2018. As a result, 41 of 195 diatom taxa were found to be the most frequent (>10% share in assemblage). According to Detrented Correspondence Analysis (DCA), three subgroups were described as the spring section, Ankara and Polatlı section, and lowland section. The river basin quality was evaluated as moderate or lower quality status, while only a few sites had good status. The diatom index scores showed that the Descy's Index (DES), Pampean Diatom Index (IDP), Artois-Picardie Diatom Index (IDAP), and Specific Pollution Sensitivity Index (IPS) appear best suited to water quality assessment in this area, showing the largest number of significantly important correlation with environmental variables.

**Keywords:** diatom indices; monitoring; ecological status; Water Framework Directive

---

## 1. Introduction

Biological and physico-chemical monitoring have been applied in order to detect the effects of human activities on aquatic environments [1]. Water quality assessment base on physicochemical analyzes is determine the water quality only at the time of measurement. This is why, such assessment is incomplete and inaccurate because water parameters may change fast over a short time. For example, in the case of uncontrolled sewage inflow. Biological monitoring allow to analyze of this constantly changing of physical and chemical characteristics of the water, giving a real reflection of conditions in

aquatic environment. One of first methods using living organisms in water quality assessment was Kolkwitz and Marsson saprobic system [2,3]. The use of diatoms in assessment of lotic and lentic waters, paleoenvironmental reconstructions, and climate studies was started in the 1970s [4,5].

The revolutionary approach in water quality assessment was introduced by Water Framework Directive [6]. Directive assumes an integrated and coordinated approach to water management in Europe based on river basin planning and monitoring. Directive also implement the concept of "ecological status" as an expression of the quality of the structure and functioning of aquatic ecosystems associated with surface waters. In the determination of ecological status, the most important factors have become the biological and hydromorphological elements. The ecological assessment of aquatic environments is based on four group of organisms: aquatic invertebrates, fishes, macrophytes, and phytobenthos microorganisms—the diatoms. The physico-chemical water parameters becomes only supporting for the biological elements [6].

The wide diatom use in environmental studies is due to fact that they show high correlation with water parameters. The diatoms are able to quickly react to changes in the water because of short life cycle. Another advantage of them is to have narrow ecological tolerance range. They are sensitive to, for example, salinity, temperature, pH, shading, water velocity, type of substratum, water chemistry and heavy metal contents. They can be easily collected and stored for a long time [7]. They are ideal organisms for water quality monitoring [8–11], and for this purpose diatom communities are used in routine monitoring programs. The diatom water quality indices were developed and designed in most cases base on regional data and for local environment quality assessment. In Europe [12–18] and United States [19,20], diatoms are widely used in water quality assessment. Most of these indices can be calculated by using Omnidia software [21] or based on ecological data available in this software [22]. Cosmopolitan distribution of diatom taxa should give comparable results of indices, however studies showing that European indices may need to be adjusted to regional conditions [12,23]. Most of the diatom indices were created for temperate climates in the Northern Hemisphere, and often cannot be applied to tropical areas or are limited to specific conditions. Recently, warmer and tropical regions have been studied by using diatom indices [24]. Important data for biological water quality assessment are publish from Mediterranean areas [25–30], which also covering the south part of Turkey.

In Turkey, the biomonitoring program has been carried out since 2011 [31] and is still developing in order to adapt to European Union legislation, especially in relation to the Water Framework Directive. Turkish waters are especially subject to this legislation and examined accordingly. The studies of Turkish inland waters mainly concerned lakes [32], while studies of flowing waters are sparse. In the Sakarya Basin, the main rivers were investigated by different researchers. The first diatom studies in the region were conducted by Atıcı and Yıldız [33,34]. A study concerning using diatoms to assess water quality was conducted by Çetin and Demir [29], mainly in the upper section of the river. A few studies concerning diatoms have also been conducted in some tributaries of the Sakarya River [32,34,35].

The Sakarya River Basin is the third-biggest river basin in Turkey, and includes highly populated cities. The present study is the first large-scale, comprehensive, scientific attempt to test the diatom indices and determine the main diatom taxa occurring in one of the most important river basins in Turkey.

## 2. Methods

### 2.1. Study Area

The Sakarya River flows through the Anatolia region (Turkey), and has a length of 824 km. The basin covers 25 major tributary basins. The catchment area is ca. 58,160 $km^2$, with an average altitude of 965 m. The climate of the region is continental. The average annual precipitation is about 480 mm [36], with a mean temperature value of 10.5 °C [37]. The main parent rock of this region is characterized by Triassic rock, composed mostly of conglomerates and sandstone which give way to Jurassic-Lower Cretaceous limestone and Upper Cretaceous flysch [37]. The springs of the Sakarya

River are located in the Eskişehir-Çifteler area. The main aquifer is composed of shelf-type carbonates of the Triassic-Upper Cretaceous periods. Dolomite limestone is dominant in the lower section, while the upper section of the unit is mostly chert limestone [38].

The Sakarya River catchment area is very important in terms of the economy, agriculture, and ecology of the country, because two highly populated cities are located in the area (the capital Ankara and Sakarya). Land use of the study area is 52.6% agricultural, forest and semi-natural areas cover 44.2%, artificial uses account for 2.5%, and 0.7% is occupied by waters and wetlands. Twenty-two percent of the agricultural areas can be irrigated, and this is an important factor for the availability and quality of surface waters [36].

A total of 46 stations were surveyed in the rivers and streams of the basin during May 2018 (Figure 1, Table S1).

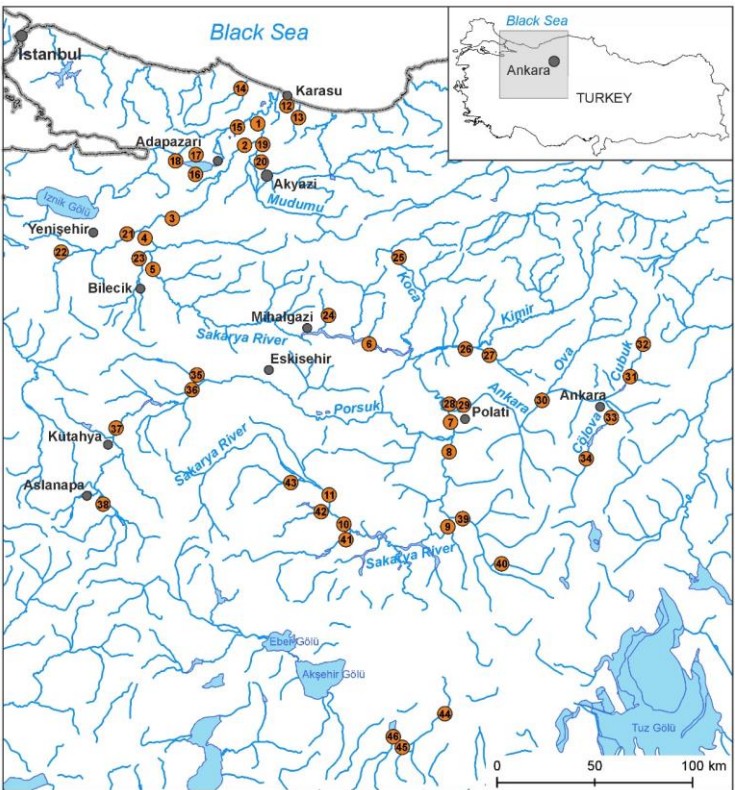

**Figure 1.** Location of the sampling stations (1–46) in the Sakarya River basin.

## 2.2. Field and Laboratory Studies

Sampling was conducted according to a standard method used in these types of studies [17,22]. The diatoms were collected by brushing submerged stones with hard brush. Samples were collected from 46 sampling stations during May 2018. In the laboratory, samples were boiled with 30% $H_2O_2$ and HCl to remove organic matter. To obtain clean diatom valves, samples were washed with distilled water in centrifuge (2500 RMP). In the next step the material was air-dried on cover glasses and mounted in Naphrax® (Brunel Microscopes Ltd, Chippenham, United Kingdom) resin. Light microscope (LM) observations were conducted using an OLYMPUS BX-51 (Mitsubishi UFJ Financial Group, Tokyo, Japan). Diatoms were identified according to literature published elsewhere [39–52].

At least 400 valves in each slide were counted for all samples. Species with a content above 10% of all counted valves in diatom assemblage were defined as the most abundant. Diatom indices were calculated using OMNIDIA 4.2 (Michel Coste, Bordeaux, France) software [21]. Due to the small number of data concerning diatoms used in water quality assessment of Turkish rivers, all available diatom indices were calculated for test their applicability (Table 1). In indices calculation, the centric

diatoms (as mainly planktonic taxa) were excluded [22]. In the present study, class limit values recommended by Elorenta and Soinien [53] and Dumnicka et al. [54] were used for the interpretation of the scores yielded by the indices. All the indices were transformed in Omnidia software to range from 0 (or 1) to 20 to be comparable (Table 2). Additionally, the Trophic Index of Turkey (TIT) was calculated for checking its applicability for assessment water quality of studied waters [55].

**Table 1.** List of diatom indices calculated for present work.

| Index | Reference | Stressor Type Sensibility |
|---|---|---|
| Artois-Picardie Diatom Index (IDAP) | [56] | General pollution |
| Eutrophication/Pollution Index (EPI-D) | [15] | Pollution/trophc status |
| Biological Diatom Index (IBD) | [57] | General pollution |
| Steinberg and Schiefele's Index (SHE) | [58,59] | Pollution/trophic status |
| Swiss Diatom Index (DI-CH) | [60] | Trophic status |
| Specific Pollution Sensitivity Index (IPS) | [61] | General pollution |
| Sládeček's Index (SLA) | [62] | Saprobity (BOD) |
| Descy's Index (DES) | [63] | General pollution |
| Louis-Leclercq Diatomic Index (IDSE) | [64] | Saprobity |
| Generic Diatom Index (IDG) | [65] | General pollution |
| Commission for Economical Community Metric—European Index (CEE) | [66] | General pollution |
| Trophic-Saprobic index (LOBO) | [67] | Eutrophication |
| Trophic Diatom Index (TDI) | [16] | Trophic status |
| Proportion of taxa tolerant to organic pollution % PT | [16] | Trophic status (Eutrophication) |
| Rott's Saprobic Metric (ROTTs) | [68] | Saprobic status |
| Rott's Trophc Metric (ROTTt) | [69] | Trophic status |
| Watanabe Index (WAT) | [70] | Saprobity (BOD) |
| Pampean Diatom Index (IDP) | [71] | Organic pollution/eutrophication |

**Table 2.** Class limit values for diatom indices according to [53,54].

| Index Score | Ecological Status | Trophy |
|---|---|---|
| >17 | high | oligotrophy |
| 15–17 | good | oligo-mesotrophy |
| 12–15 | moderate | mesotrophy |
| 9–12 | poor | meso-eutrophy |
| <9 | bad | eutrophy |

Water temperature, dissolved oxygen (DO), pH and electrical conductivity (EC) were measured in situ using portable equipment (Lange Hach 40d). For detailed chemical analysis, the water samples were taken from the main flow of the watercourses. The samples were stored in iceboxes and transferred to the laboratory for analysis. Total nitrogen (TN), ammonium nitrogen ($NH_4^+$), nitrite ($NO_2^-$), nitrate ($NO_3^-$), total phosphorus (TP), and orthophosphate ($PO_4^{3-}$) were determined according to APHA [72].

In order to verify which index had the strongest correlation with environmental variables, the Pearson correlation was performed by Statistica 13.3. Statistically significant data were considered for *p*-value < 0.05 and <0.01.

To determine the diatom assemblages' similarity, the Detrented Correspondence Analysis (DCA) was made (gradient length: 4.1) with downweight rare species option. The eigenvalues were 0.389, 0.228, 0.149 and 0.117 respectively for each axes.

The Canonical Correspondence Analysis (CCA) was applied to analyze the influence of environmental factors on diatom assemblages. The gradient length for the CCA was 6.4 SD. Significant test performed for CCA analysis did not show statistical importance (*p* = 0.122).

Both analyses (DCA and CCA) were performed using Canoco 5 software.

## 3. Results

### 3.1. Physico-Chemical Water Parameters

The water chemistry was highly changeable, depending on the part of the drainage area. Water temperature ranged between 12.7 and 23 °C, and pH was alkaline at each station (7.2–8.4).

Electrolytic conductivity was highly variable, from 189 μS·cm$^{-1}$ (station 13) to 5 910 μS·cm$^{-1}$ (station 7). The lowest values were noted mainly in the upper sections of Sakarya River tributaries. The highest values of conductivity and nutrient levels (nitrate and phosphate ions, total nitrogen, and total phosphate) and total organic carbon content were noted at stations located in the Ankara and Polatlı areas. The content of total nitrogen and forms of phosphorus were especially high at station 33 located near the capital city, Ankara, and at station 29 near Polatlı in district of Ankara. In these locations (i.e., stations 28, 29, 33) the levels of ammonium content were also very high (4.03–12.0 mg·L$^{-1}$). At other sampling sites, the phosphate content was generally low (0.02–0.92 mg·L$^{-1}$). In general, the lowest values of nutrients, especially in terms of total nitrogen content, was noted at stations located in the upper sections of the Sakarya River (stations 10, 11, 43) and its tributaries (i.e., at Porsuk River—stations 37, 38). The biochemical oxygen demand was extremely changeable, from 1 to 80 mg O$_2$·L$^{-1}$. The highest BOD$_5$ levels were noted in the same stations at Polatlı and Ankara, while the lowest in the small tributaries of the Sakarya River. The highest nutrient, total organic carbon, COD and BOD$_5$ values were also noted at station 22, located around Yenişehir district in Ankara, and at stations 15 and 19 in the lower section of the Sakarya River basin (Table 3).

**Table 3.** The physico-chemical parameters of water studied: Temp.—water temperature, EC—electrolytic conductivity, DO—dissolved oxygen, BOD—biochemical oxygen demand, COD—chemical oxygen demand, TOC—total organic carbon, TP—total phosphate.

| Station Number | Temp. °C | pH | EC $\mu S \cdot cm^{-1}$ | DO $mg \cdot L^{-1}$ | BOD mg $O_2 \cdot L^{-1}$ | COD mg $O_2 \cdot L^{-1}$ | TOC $mg \cdot L^{-1}$ | TN $mg \cdot L^{-1}$ | $NH_4^+$ $mg \cdot L^{-1}$ | $NO_{2-}$ $mg \cdot L^{-1}$ | $NO_{3-}$ $mg \cdot L^{-1}$ | TP $mg \cdot L^{-1}$ | $PO_4^{3-}$ $mg \cdot L^{-1}$ |
|---|---|---|---|---|---|---|---|---|---|---|---|---|---|
| 1 | 20 | 8 | 620 | 8 | 4 | 7 | 3.93 | 2.26 | 0.078 | 0.1 | 1.43 | 0.22 | 0.238 |
| 2 | 18.7 | 7.7 | 568 | 7.3 | 3 | 9.8 | 3.52 | 1.56 | 0.078 | 0.077 | 0.89 | 0.44 | 0.646 |
| 3 | 17.4 | 8 | 540 | 8.8 | 1 | 14.4 | 3.99 | 1.11 | 0.06 | 0.07 | 1.41 | 0.17 | 0.308 |
| 4 | 18.2 | 8.1 | 667 | 8.7 | 1 | – | – | 2.2 | 0.05 | – | 1.414 | 0.11 | – |
| 5 | 18 | 8.2 | 818 | 8.5 | 2 | 12.4 | 3.96 | 2.31 | – | 0.13 | 1.56 | – | 0.693 |
| 6 | 15.8 | 8.4 | 1037 | 9.2 | 4 | 15.9 | 3.61 | 8.95 | 0.731 | <0.018 | 5.116 | 0.607 | 1.638 |
| 7 | 20.7 | 8.3 | 5910 | 11.3 | 30 | 35.3 | 2.46 | 10.71 | <0.047 | <0.018 | 8.482 | 0.467 | 1.236 |
| 8 | 19.2 | 8.1 | 2059 | 6 | 9 | 11.4 | 3.09 | 10.13 | <0.047 | <0.018 | 7.972 | 3.029 | 0.892 |
| 9 | 19.9 | 8.1 | 1570 | 7.6 | 4 | 6.9 | 1.76 | 2.66 | <0.047 | <0.018 | 1.723 | 0.032 | 0.057 |
| 10 | 21.1 | 8.2 | 1120 | 8.6 | 1 | – | – | 0.97 | – | – | 0.59 | 0.11 | – |
| 11 | 19.6 | 8.1 | 910 | 8.1 | 1 | 5.7 | 2.63 | 0.79 | – | 0.08 | 0.505 | 0.21 | 0.401 |
| 12 | 22.3 | 8.2 | 1637 | 13.8 | 3 | – | – | 0.57 | – | – | 0.212 | 0.05 | – |
| 13 | 16.9 | 8.1 | 189 | 9.8 | 1 | – | – | 2.17 | – | – | 0.73 | 0.04 | – |
| 14 | 20.1 | 8 | 571 | 8.1 | 1 | 9.1 | – | 4.01 | 0.1 | – | 2.01 | 0.18 | 0.39 |
| 15 | 17.5 | 7.6 | 693 | 4.1 | 9 | 23 | 8.71 | 6.07 | 2.71 | 0.48 | 1.079 | 1.66 | 1.258 |
| 16 | 16.5 | 8 | 722 | 9.9 | 3 | 3.8 | 2.27 | 0.44 | – | – | – | – | – |
| 17 | 19.2 | 8.1 | 416 | 8.9 | 2 | 3.2 | 1.68 | 0.78 | – | – | 0.54 | 0.24 | 0.05 |
| 18 | 16.9 | 7.2 | 311 | 4.2 | 5 | 16.1 | 7.39 | 1.24 | 0.271 | 0.159 | 0.457 | 0.358 | <0.260 |
| 19 | 16.6 | 8.1 | 376 | 9 | 2 | – | – | 7.98 | 0.046 | – | 1.45 | 0.06 | – |
| 20 | 17.6 | 7.9 | 372 | 8.5 | 2 | – | – | 1.87 | 0.09 | – | 1.49 | 0.05 | – |
| 21 | 16.4 | 8.2 | 346 | 10.2 | 3 | 17.8 | 4.81 | 2.48 | 0.008 | 0.09 | 1.07 | – | 0.281 |
| 22 | 16.3 | 7.7 | 593 | 3.8 | 9 | 23.6 | 9.86 | 3.34 | 0.661 | 1.128 | 8.243 | 0.827 | 0.863 |
| 23 | 16.2 | 8.2 | 405 | 8.7 | 4 | 18.8 | 6.39 | 3.74 | 0.778 | 0.22 | 1.46 | – | 0.36 |
| 24 | 17.2 | 8.5 | 510 | 8.9 | 1 | <5.00 | 1.52 | 1.29 | <0.047 | <0.018 | 0.895 | 0.019 | 0.02 |
| 25 | 13.9 | 7.5 | 300 | 9.6 | 4 | 4.9 | 3.34 | 1.1 | <0.047 | <0.018 | 0.635 | 0.051 | 0.095 |
| 26 | 17.2 | 8.5 | 387 | 10.8 | 9 | 10.5 | 4.22 | 1.56 | 0.227 | 0.021 | 1.281 | 0.103 | 0.266 |
| 27 | 17 | 7.9 | 2790 | 9.7 | 1 | – | – | 10.81 | – | – | 9.816 | 0.091 | 0.186 |
| 28 | 23 | 7.7 | 1294 | 4.1 | 31 | – | – | 13.24 | 15.14 | – | – | 1.97 | – |
| 29 | 21.8 | 7.8 | 1426 | 2.2 | 70 | 106.9 | 8.11 | 19.95 | 11.96 | <0.018 | 1.833 | 2.056 | 5.703 |
| 30 | 19.9 | 7.8 | 1379 | 0.6 | 80 | 118.7 | 7.2 | 8.9 | 5.01 | <0.018 | 2.287 | 0.742 | 1.928 |
| 31 | 17.2 | 8.2 | 503 | 7.6 | 1 | 9.2 | 3.07 | 3.36 | – | – | 3.07 | 0.58 | 0.923 |

**Table 3.** *Cont.*

| Station Number | Temp. °C | pH | EC μS·cm$^{-1}$ | DO mg·L$^{-1}$ | BOD mg O$_2$·L$^{-1}$ | COD mg O$_2$·L$^{-1}$ | TOC mg·L$^{-1}$ | TN mg·L$^{-1}$ | NH$_4^+$ mg·L$^{-1}$ | NO$_2$$_-$ mg·L$^{-1}$ | NO$_3$$_-$ mg·L$^{-1}$ | TP mg·L$^{-1}$ | PO$_4^{3-}$ mg·L$^{-1}$ |
|---|---|---|---|---|---|---|---|---|---|---|---|---|---|
| 32 | 17.5 | 7.9 | 907 | 11.2 | 4 | – | – | 2.48 | – | – | 0.871 | 0.142 | 0.345 |
| 33 | 18.9 | 7.6 | 1269 | 2.3 | – | – | – | 30.31 | 4.028 | – | 1.275 | 1.176 | 3.195 |
| 34 | 19.8 | 7.9 | 1021 | 6.2 | <1 | 18 | 5.66 | 3.29 | <0.047 | <0.018 | 1.273 | 0.034 | 0.069 |
| 35 | 16 | 7.9 | 466 | 7.3 | 2 | 15.4 | 6.36 | 1.596 | 0.33 | 0.086 | 0.79 | 0.36 | 0.495 |
| 36 | 12.7 | 8.1 | 481 | 9.1 | 5 | 11.4 | 4.01 | 1.52 | – | 0.025 | 1.034 | 0.11 | 0.181 |
| 37 | 17.4 | 7.7 | 592 | 3.5 | 3 | 20.4 | 4.77 | 1.22 | 0.12 | 0.28 | 0.045 | 0.22 | 0.108 |
| 38 | 18 | 8.4 | 489 | 12.1 | 4 | 11 | 3.41 | 0.35 | – | 0.037 | 0.17 | 0.07 | 0.107 |
| 39 | 18.9 | 8 | 1633 | 8.9 | 4 | 6.4 | 1.42 | 2.84 | <0.047 | <0.018 | 2.181 | 0.029 | 0.045 |
| 40 | 23 | 8.4 | 1297 | 10.6 | 7 | – | – | 6.5 | 0.741 | – | 5.579 | 0.076 | 0.146 |
| 41 | 17.3 | 8 | 1182 | 7.7 | 1 | – | – | 2.29 | – | – | 1.82 | 0.54 | – |
| 42 | 21.7 | 8.2 | 1065 | 8.9 | 1 | 5.8 | 3.06 | 1.42 | – | 0.06 | 1.03 | 0.33 | 0.14 |
| 43 | 17 | 7.5 | 802 | 3.6 | 1 | 6.5 | 1.42 | 0.68 | – | 0.06 | 0.276 | 0.16 | 0.355 |
| 44 | – | – | – | – | – | – | – | – | – | – | – | – | – |
| 45 | 17.5 | 8.1 | 413 | 7.5 | 4 | 19.9 | 4.63 | 1.25 | <0.100 | <0.100 | 4.702 | 0.22 | – |
| 46 | 19.2 | 8.4 | 362 | 7.2 | 4 | – | – | 0.28 | – | – | – | 0.074 | – |

### 3.2. Diatom Composition

A total of 195 diatom taxa belonging to 67 genera were identified at the sampling stations of the Sakarya Basin (Table 4). Excluding centric diatoms (*Cyclotella meneghiniana* Kützing and *Stephanodiscus neoastraea* Håkansson and Hickel), which are not taken into consideration for diatom indices calculation, 41 diatom taxa were considered as the most abundant.

**Table 4.** Diatom taxa recorded during studies.

| | | |
|---|---|---|
| *Achnanthidium pyrenaicum* | *F. pygmaea* | *Neidiomorpha binodiformis* |
| *Adlafia minuscula* | *Fistulifera saprophila* | *Nitzschia acicularis* |
| *A. minuscula* var. *muralis* | *Fragilaria famelica* | *N. amphibia* |
| *Amphipleura pellucida* | *F. tenera* | *N. archibaldii* |
| *Amphora copulata* | *Frustulia vulgaris* | *N. capitellata* |
| *A. inariensis* | *Geissleria decussis* | *N. clausii* |
| *A. ovalis* | *Gomphonema calcifugum* | *N. communis* |
| *A. pediculus* | *G. exillissimum* | *N. dissipata* |
| *Aneumastus minor* | *G. italicum* | *N. dubia* |
| *Anomoeoneis sphaerophora* | *G. minutum* | *N. filiformis* |
| *Asterionella formosa* | *G. olivaceum* | *N. fonticola* |
| *Aulacoseira ambigua* | *G. parvulum* | *N. frustulum* |
| *A. granulata* var. *angustissima* | *G. subclavatum* | *N. hantzschiana* |
| *Bacillaria paxillifera* | *Gi tergestinum* | *N. heufleriana* |
| *Brachysira procera* | *G. truncatum* | *N. inconspicua* |
| *Caloneis amphisbaena* | *Gyrosigma attenuatum* | *N. intermedia* |
| *C. lancettula* | *G. kuetzingii* | *N. linearis* |
| *C. silicula* | *G. obtusatum* | *N. media* |
| *Cocconeis pediculus* | *G. sciotense* | *N. microcephala* |
| *C. placentula* | *Halamphora montana* | *N. palea* |
| *C. placentula* var. *lineata* | *H. veneta* | *N. pusilla* |
| *C. pseudolineata* | *Hantzschia amphioxys* | *N. radicula* |
| *Conticribra weissflogii* | *Hippodonta capitata* | *N. recta* |
| *Craticula accomoda* | *Humidophila contenta* | *N. sociabilis* |
| *C. ambigua* | *Karayevia clevei* | *N. solita* |
| *C. buderi* | *K. pleonensis* | *N. subtilis* |
| *C. molesta* | *Lemnicola hungarica* | *N. thermaloides* |
| *C. subminuscula* | *Lindavia balatonis* | *N. umbonata* |
| *Ctenophora pulchella* | *Luticola mutica* | *N. wuellerstroffii* |
| *Cyclostephanos dubius* | *L. nivalis* | *Pantocsekiella ocellata* |
| *C. invisitatus* | *L. ventricosa* | *Pinnularia brebissonii* |
| *Cyclotella atomus* | *L. similis* | *Planothidium lanceolatum* |
| *C. cryptica* | *Mayamaea atomus* | *Pseudostaurosira brevistriata* |
| *C. meneghiniana* | *Melosira varians* | *Reimeria sinuata* |
| *Cymatopleura solea* | *Meridion circulare* | *R. uniseriata* |
| *C. solea* var. *apiculata* | *Navicula antonii* | *Rhoicosphenia abbreviata* |
| *Cymbella compacta* | *N. capitatoradiata* | *Rhopalodia gibba* |
| *C. excisa* | *N. cari* | *Sellaphora pupula* |
| *C. neocistula* | *N. caterva* | *S. radiosa* |
| *C. neolanceolata* | *N. cincta* | *S. seminulum* |
| *C. tumida* | *N. cryptocephala* | *S. saugerressii* |
| *Cymbopleura amphicephala* | *N. cryptofallax* | *Stauroneis separanda* |
| *C. vrana* | *N. cryptotenella* | *S. smithii* |
| *Denticula kutzingii* | *N. cryptotenelloides* | *Staurophora tackei* |
| *Diadesmis confervaceae* | *N. erifuga* | *Staurosira construens* |
| *Diatoma ehrenbergii* | *N. germainii* | *Stephanodiscus hantzschii* |
| *D. mesodon* | *N. gottlandica* | *S. minutulus* |
| *D. moniliformis* | *N. gregaria* | *S. neoastreae* |
| *D. tenuis* | *N. kotschyi* | *Surirella angusta* |
| *D. vulgaris* | *N. lacuum* | *S. brebissonii* |

**Table 4.** *Cont.*

| | | |
|---|---|---|
| *Diploneis oculata* | *N. lanceolata* | *S. gracilis* |
| *D. separanda* | *N. novaesiberica* | *S. minuta* |
| *Discostella stelligera* | *N. oblonga* | *S. neglecta* |
| *Ellerbeckia arenaria* | *N. phylleptosoma* | *S. ovalis* |
| *Encyonema caespitosum* | *N. radiosa* | *Tabularia fasciculata* |
| *E. lacustre* | *N. reichardtiana* | *Tryblionella angustata* |
| *E. minutum* | *N. rostellata* | *T. angustatula* |
| *E. silesiacum* | *N. salinarum* | *T. apiculata* |
| *E. ventricosum* | *N. simulata* | *T. brunoi* |
| *Encyonopsis minuta* | *N. tripunctata* | *T. calida* |
| *E. subminuta* | *N. trivialis* | *T. debilis* |
| *Entomoneis paludosa* var. *subsalina* | *N. upsaliensis* | *T. hungarica* |
| *Epithemia adnata* | *N. vandamii* | *Ulnaria acus* |
| *E. sorex* | *N. vilaplanii* | *U. biceps* |
| *Fallacia lenzii* | *N. veneta* | *U. nanana* |

The most common taxa in the basin are listed in Figure 2. The most frequent taxa in the Sakarya River were *Amphora pediculus* (Kützing) Grunow, *Cymbella excisa* Krammer, *Cyclotella meneghiniana* Kützing, and *Stephanodiscus neoastraea* (Kützing) Grunow, while *Navicula lanceolata* Ehrenberg, *Nitzschia dissipata* (Kützing) Rabenhorst, and *Stephanodiscus neoastraea* (Kützing) Grunow were dominant in the Porsuk River—another important river and one of the main branches of the Sakarya River. Other important rivers in the basin are the Ankara and Çubuk Rivers. *Craticula accomoda* (Hustedt) D.G.Mann, *Craticula subminuscula* (Manguin) C.E.Wetzel and Ector and *Fistulifera saprophila* (Lange-Bertalot and Bonik) Lange-Bertalot were abundant taxa in these rivers.

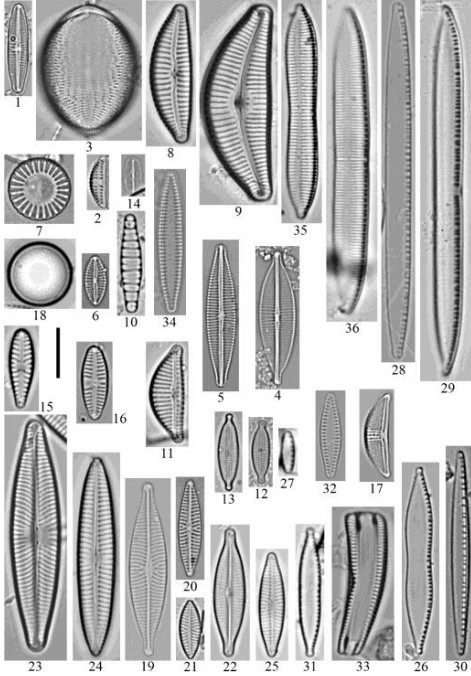

**Figure 2.** The most common taxa in the basin and abundant taxa in the stations. 1—*Achnanthidium pyrenaicum*, 2—*Amphora pediculus*, 3—*Cocconeis pediculus*, 4—*Craticula accomoda*, 5—*Craticula buderi*, 6—*Craticula subminuscula*, 7—*Cyclotella meneghiniana*, 8—*Cymbella excisa*, 9—*C. neocistula*, 10—*Diatomamoniliformis*, 11—*Encyonema ventricosum*, 12—*Encyonopsis minuta*, 13—*E. subminuta*, 14—*Fistulifera saprophila*, 15—*Gomphonema olivaceum*, 16—*Gomphonema tergestinum*, 17—*Halamphora veneta*, 18—*Melosira varians*, 19—*Navicula capitatoradiata*, 20—*N. cryptotenella*, 21—*N. cryptotenelloides*,

22—*N. gregaria*, 23—*N. lanceolata*, 24—*N. tripunctata*, 25—*N. veneta*, 26—*Nitzschia capitellata*, 27—*N. inconspicua*, 28—*Nitzschia intermedia*, 29—*N. linearis*, 30—*N. media*, 31—*N. palea*, 32—*Pseudostaurosira brevistriata*, 33—*Rhoicosphenia abbreviata*, 34—*Tabularia fasciculata*, 35—*Tryblionella apiculata*, 36—*T. hungarica*. Scale bar: 10 µm.

For the study of the similarities in diatom assemblages, the DCA analysis was performed for all taxa identified at each site. The DCA analysis showed the differentiation of diatom assemblages mainly by location of the sampling sites in the area of the basin (Figure 3). The diatom community structure is closely reflected by a gradient of increasing pollution, from highly polluted sites to other communities. Three groups were observed as a result of the DCA analysis. The first (group A) includes diatom assemblages collected from the station located mainly in the spring section of the Sakarya River and small tributaries (Figure 3). This association was dominated by *Achnanthidium pyrenaicum* (Hustedt) Kobayasi, *Amphora pediculus*, *Cymbella excisa*, *Gomphonema tergestinum* (Grunow) Fricke, *Navicula cryptotenelloides* Lange-Bertalot, *Pseudostaurosira brevistriata* (Grunow) D.M. Williams and Round, *Rhoicosphenia abbreviata* (Grunow) D.M. Williams and Round and *Staurosira construens* Ehrenberg. The second group (group B) consisted of assemblages from areas around Ankara and Polatlı (Figure 3), and was dominated by *Craticula accomoda*, *C. buderi* (Hustedt) Lange-Bertalot, *C. subminuscula*, *Fistulifera saprophila*, *Navicula cryptotenella* Lange-Bertalot, *Nitzschia capitellata* Hustedt, *N. palea* (Kützing) W. Smith, *N.pusilla* Grunow and *Tryblionella hungarica* (Grunow) Frenguelli. The third group (group C) consisted of assemblages mainly from the stations located in the lower section of the Sakarya River and its tributaries (Figure 3). The most frequent taxa to occur in this group were *Amphora pediculus*, *Navicula tripunctata* (O.F.Müller) Bory, *N. lanceolata*, *N. gregaria* Donkin, *Nitzschia palea*, *N. inconspicua* Grunow, and less frequently, *Cocconeis placentula* Ehrenberg, *Fistulifera saprophila*, *Navicula caterva* Hohn and Hellermann, *N. cryptotenella*, *N. veneta* Kützing, *Nitzschia dissipata*, *N. intermedia* Hantzsch, *N. media* Hantzsch, and *Ulnaria nanana* Lange-Bertalot. The most distinctive assemblage was found in station 34 (Figure 3). It was the only station dominated by *Ctenophora pulchella* (Ralfs ex Kützing) D. M. Williams and Round, *Halamphora veneta* (Kützing) Levkov, and *Tabularia fasciculata* (C. Agardh) D. M. Williams.

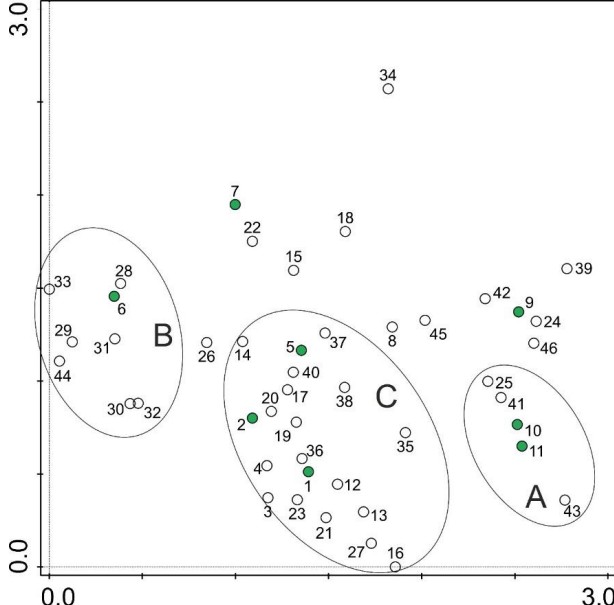

**Figure 3.** Results of the differentiation of diatom assemblages based on Detrented Correspondence Analysis (1–46—sampling stations, **A**–**C**—groups of diatom assemblages: A—spring section of the Sakarya River and small tributaries, B—assemblages from the area around Ankara and Polatlı, C—lower section of the Sakarya River and tributaries, green dots correspond to stations located on the Sakarya River).

To examine the influence of the environmental variables on the most abundant diatom taxa, a CCA analysis was performed. The altitude of stations was significant ($p = 0.008$) and explains 9.0% of the variability of diatom assemblages, but the significance test performed for CCA analysis did not show statistical importance ($p = 0.122$).

The diatom index scores in the sampling stations are given in Table 5. Index values were different through the stations according to the percentage of species used in the calculation of indices. The IBD, IPS, IDG, and TDI indices included the highest amount of species diversity. The IBD, IPS, and IDG indices were calculated in almost all stations with more than 90% identified taxa. The DES and LOBO indices worked in a few stations with only about 60% of our species. Regarding IBD, 7% of the stations were of "high" and "good" quality status, while 93% were "moderate" or lower. For IPS, 19% of the stations were "good" and 81% were "moderate" or lower. For IDG, only 8% of the stations were "high" or "good" and 92% were "moderate" or lower ecological status.

As a result, the river basin quality was evaluated as moderate or lower quality status, while only a few sites had good status (Table 5). The Sakarya River generally had moderate or lower water quality status according to IPS and IDG indices, while a few sites were of good status regarding the IBD index. The Porsuk River have bad quality below Kütahya province. The main reason for the low water quality was related to certain regional industrial discharges, such as from ceramic factories. Based on the index results, the Ankara River has a relatively better status than the Sakarya and Porsuk Rivers. The diatom index values were highly variable along the Sakarya River. Higher values were noted in spring sections of rivers and streams (i.e., group A) (Figure 3), and in the lower section at stations 1, 3, 5, 8, and 9 located near Polatlı. At certain Sakarya River tributaries high values of indices were recorded in the spring section of the Porsuk River (station 38) and small watercourses in the north part of the basin (stations 12, 13). The lowest values of diatom indices were recorded in stations located in an area of Polatlı and Ankara (i.e., group B) (Figure 3), in the Porsuk River (station 37), and also at stations 40, 44, and 45. The the highest values (the highest ecological status) were obtained by the DES (mean = 13.1), WAT (mean = 11.3), LOBO (mean = 11.1), ROTT saprobic (mean = 10.7), SHE (mean = 10.5), and IPS (mean = 10.1) indices. The lowest ecological status was determined by the TDI index (mean = 4.7) and ROTT trophic (mean = 5.4). The water typology is an integration component of the TIT (Trophc Index Turkey) formula. However, since determination of the ecoregions (water typology) of the Sakarya River Basin is not complete, the ecological status according to the TIT cannot be determined in the present study.

The correlations between all the diatom indices, including the Shannon Diversity (H') and Evenness indices, physicochemical parameters and altitude, were evaluated at *p*-values 0.05 and 0.01. The IDAP, EPI-D, IBD, SHE, DI-CH, IPS, SLA, DES, IDSE, and ROTT saprobic indices had negative significant correlations with $BOD_5$, COD, Total-N, $NH_4^+$, and $PO_4^3$ at *p*-value 0.05 (the correlation coefficient ranged between 0.45 and 0.65). The IBD and DES indices also had positive significant correlations with DO (i.e., r = 0.39 and 0.44 respectively) and only the DES index had negative significant correlations with Total P (r = −0.44) at $p = 0.01$. Significant positive correlations were also determined between Dissolved Oxygen and IDAP, SHE, IPS, SLA, IDSE, and IDP indices at *p*-value 0.05 (correlation coefficients ranged between −0.31 and −0.34). Altitude, Temperature, TOC and Total P mostly showed negative correlations with the indices, and some were significant at *p*-value 0.05 (Table 6). Among the indices evaluated, IDP, DES, and IDAP were significantly correlated with the largest number of environmental variables (i.e., 11, 11, and 10, variables respectively). The CEE, ROTT trophic, TDI, and IDG indices had 4, 3, 1, and 1 significant correlations with the environmental variables at *p*-value 0.01, and had 3, 4, 1, and 2 significant correlations at *p*-value 0.05, respectively. The WAT, TIT, and IDP indices had no significant correlations with the environmental variables at *p*-value 0.01, whereas they had 6, 2 and 11 significant correlations with the same variables at *p*-value 0.05, respectively. On the other hand, the LOBO and % PT indices had no significant correlations with the examined variables. In general, most of diatom indices evaluated in the frame of the present study were significantly correlated with important nutrients (Table 6).

**Table 5.** The diatom index values at sampling stations and corresponding ecological status (blue—high, green—good, yellow—moderate, orange—poor, red—bad).

| Station No. | IDAP | EPI-D | IBD | SHE | DI-CH | WAT | IPS | SLA | DES | IDSE/5 | IDG | CEE | TDI/20 | %PT | LOBO | IDP | ROTTt | ROTTs | TIT | H′ | Evenness |
|---|---|---|---|---|---|---|---|---|---|---|---|---|---|---|---|---|---|---|---|---|---|
| 1 | 14.2 | 11.2 | 12.5 | 13.7 | 11.3 | 12.3 | 15.7 | 11.9 | 19.5 | 13.6 | 12.3 | 12.8 | 2.9 | 18.2 | 16.3 | 10.6 | 5.3 | 13.0 | 3.1 | 2.92 | 0.65 |
| 2 | 8.7 | 7.4 | 4.5 | 9.1 | 7.2 | 10.7 | 6.7 | 9.1 | 17.2 | 9.3 | 9.1 |  | 3.7 | 12.3 | 11.8 | 9.1 | 4.5 | 10.5 | 3.2 | 2.28 | 0.47 |
| 3 | 12.4 | 9.7 | 10.7 | 12.7 | 9.2 | 11.8 | 14.0 | 10.8 | 18.8 | 12.8 | 10.3 | 12.2 | 2.5 | 29.3 | 16.4 | 9.6 | 4.8 | 12.3 | 3.7 | 2.82 | 0.85 |
| 4 | 10.8 | 7.2 | 10.3 | 8.6 | 6.3 | 11.5 | 9.6 | 9.8 | 16.5 | 9.5 | 8.2 | 11.1 | 1.8 | 16.8 | 11.3 | 8.5 | 5.5 | 9.2 | 2.7 | 3.31 | 0.72 |
| 5 | 12.9 | 8.7 | 7.9 | 2.9 | 6.8 | 12.4 | 11.5 | 8.9 | 17.0 | 9.0 | 10.2 | 9.6 | 4.7 | 14.6 | 11.6 | 10.3 | 6.5 | 13.3 | 3.0 | 3.59 | 0.75 |
| 6 | 5.9 | 0.3 | 4.8 | 3.3 | 3.9 | 9.0 | 4.1 | 6.6 | 5.4 | 6.4 | 6.0 | 3.1 | 2.4 | 41.4 | 6.0 | 6.9 | 2.2 | 5.7 | 2.8 | 2.55 | 0.69 |
| 7 | 6.4 | 6.1 | 5.9 | 4.4 | 3.3 | 6.4 | 3.8 | 7.7 | 2.0 | 6.9 | 3.2 | 3.9 | 3.1 | 60.5 | 20.0 | 2.2 | 4.0 | 6.4 | 2.8 | 2.07 | 0.74 |
| 8 | 13.7 | 10.5 | 9.9 | 14.5 | 8.5 | 16.9 | 12.8 | 10.8 | 15.1 | 12.0 | 10.9 | 11.5 | 4.2 | 15.9 | 14.6 | 6.2 | 5.9 | 12.1 | 2.9 | 2.80 | 0.63 |
| 9 | 14.1 | 11.8 | 10.3 | 14.9 | 9.4 | 16.3 | 14.7 | 11.1 | 15.5 | 12.0 | 12.5 | 12.2 | 6.7 | 9.4 | 19.4 | 10.5 | 6.4 | 12.6 | 2.9 | 2.99 | 0.65 |
| 10 | 12.1 | 15.1 | 16.4 | 16.2 | 16.0 | 14.6 | 16.4 | 10.1 | 16.1 | 12.4 | 15.3 | 15.4 | 11.1 | 7.9 | 10.5 | 11.2 | 8.7 | 15.4 | 3.0 | 3.36 | 0.68 |
| 11 | 14.2 | 14.3 | 11.9 | 15.9 | 13.6 | 12.9 | 14.3 | 11.9 | 19.2 | 13.1 | 13.4 | 13.9 | 8.1 | 5.3 | 11.8 | 10.6 | 6.7 | 15.7 | 2.7 | 2.99 | 0.59 |
| 12 | 7.9 | 7.3 | 8.3 | 9.8 | 7.5 | 12.3 | 6.8 | 11.3 | 14.5 | 9.3 | 5.5 | 10.3 | 3.1 | 73.5 | 9.9 | 6.9 | 3.0 | 11.0 | 3.4 | 3.29 | 0.69 |
| 13 | 13.2 | 12.2 | 13.8 | 13.3 | 13.0 | 16.7 | 15.6 | 11.8 | 18.7 | 12.8 | 10.8 | 14.7 | 4.9 | 17.5 | 12.1 | 10.2 | 6.6 | 13.3 | 3.6 | 2.90 | 0.65 |
| 14 | 11.7 | 11.5 | 18.4 | 9.1 | 7.4 | 17.4 | 13.6 | 11.5 | 17.0 | 10.2 | 12.1 | 11.8 | 8.8 | 5.8 | 2.1 | 10.3 | 7.9 | 10.8 | 3.5 | 2.02 | 0.44 |
| 15 | 10.1 | 8.6 | 10.6 | 11.5 | 7.9 | 11.9 | 10.3 | 9.4 | 13.6 | 10.1 | 10.5 | 9.4 | 4.8 | 41.7 | 12.4 | 8.6 | 3.9 | 10.9 | 2.8 | 4.40 | 0.83 |
| 16 | 13.1 | 14.2 | 15.4 | 14.2 | 13.9 | 17.8 | 16.6 | 13.2 | 18.2 | 13.6 | 10.9 | 13.9 | 6.0 | 10.7 | 18.9 | 13.9 | 8.4 | 14.0 | 3.9 | 2.39 | 0.54 |
| 17 | 10.3 | 7.1 | 8.4 | 8.7 | 5.0 | 8.1 | 5.9 | 8.8 | 8.4 | 8.2 | 6.4 | 4.8 | 2.8 | 50.8 | 17.9 | 4.7 | 4.3 | 8.1 | 3.0 | 3.23 | 0.65 |
| 18 | 9.5 | 7.0 | 8.8 | 11.4 | 8.5 | 13.6 | 10.0 | 9.7 | 15.9 | 10.5 | 11.5 | 11.6 | 7.2 | 10.1 | 6.3 | 9.1 | 5.0 | 11.1 | 2.7 | 3.42 | 0.68 |
| 19 | 13.3 | 10.0 | 11.5 | 11.9 | 9.4 | 14.1 | 13.1 | 11.3 | 18.9 | 11.2 | 10.2 | 12.2 | 2.6 | 25.1 | 9.4 | 9.8 | 6.1 | 12.1 | 2.9 | 3.78 | 0.75 |
| 20 | 11.5 | 8.1 | 11.9 | 9.8 | 8.1 | 13.0 | 11.2 | 9.2 | 16.8 | 10.5 | 10.8 | 10.9 | 3.2 | 26.0 | 10.5 | 9.2 | 4.9 | 10.5 | 2.5 | 3.99 | 0.83 |
| 21 | 13.1 | 10.8 | 10.8 | 12.8 | 9.1 | 12.9 | 14.9 | 13.0 | 18.5 | 13.1 | 8.4 | 13.5 | 2.8 | 25.7 | 13.7 | 10.8 | 5.7 | 12.5 | 3.6 | 2.66 | 0.68 |
| 22 | 10.1 | 8.5 | 10.1 | 8.5 | 5.3 | 14.5 | 10.3 | 9.5 | 12.2 | 9.8 | 12.8 | 9.4 | 2.5 | 47.7 | 10.2 | 5.7 | 4.4 | 9.4 | 2.8 | 3.12 | 0.68 |
| 23 | 12.4 | 10.5 | 9.5 | 12.4 | 7.8 | 17.1 | 12.6 | 12.3 | 18.9 | 12.8 | 7.2 | 11.8 | 0.9 | 20.4 | 1.8 |  | 6.1 | 11.8 | 3.9 | 1.38 | 0.36 |
| 24 | 14.3 | 14.9 | 14.0 | 18.8 | 16.7 | 10.7 | 15.6 | 10.2 | 16.7 | 12.0 | 16.9 | 11.6 | 8.0 | 0.2 | 2.1 | 13.0 | 10.8 | 13.7 | 3.1 | 2.36 | 0.62 |
| 25 | 15.0 | 14.1 | 11.7 | 19.3 | 15.7 | 11.5 | 15.4 | 9.8 | 18.1 | 14.2 | 13.9 | 14.3 | 8.5 | 0.3 | 11.3 | 8.1 | 12.6 | 14.1 | 3.0 | 1.67 | 0.44 |
| 26 | 3.7 | 1.7 | 2.6 | 2.5 | 3.5 | 9.4 | 2.1 | 7.5 | 5.7 | 4.8 | 3.0 | 3.7 | 4.3 | 75.1 | 19.1 | 7.8 | 1.7 | 5.3 | 3.9 | 1.59 | 0.42 |
| 27 | 10.5 | 9.6 | 8.2 | 11.8 | 8.3 | 14.6 | 12.7 | 13.6 | 16.8 | 11.3 | 4.9 | 11.5 | 0.8 | 55.2 | 2.4 | 13.6 | 6.1 | 12.1 | 3.9 | 1.73 | 0.55 |
| 28 | 1.9 | 1.8 | 3.7 | 1.4 | 3.2 | 6.2 | 1.2 | 7.6 | 1.2 | 5.4 | 1.8 | 3.7 | 3.4 | 90.1 | 19.6 | 2.3 | 2.4 | 3.8 | 3.1 | 1.62 | 0.45 |
| 29 | 1.7 | 2.4 | 1.2 | 2.3 | 1.8 | 2.6 | 1.7 | 5.0 | 1.3 | 1.6 | 7.9 | 4.2 | 1.1 | 7.1 | 3.1 | 4.5 | 1.7 | 3.5 | 3.9 | 1.59 | 0.53 |
| 30 | 2.8 | 2.9 | 2.9 | 5.1 | 2.4 | 3.2 | 2.8 | 4.6 | 1.0 | 4.2 | 8.0 | 5.6 | 0.7 | 0.5 | 15.2 | 4.6 | 1.8 | 6.2 | 3.9 | 1.40 | 0.54 |
| 31 | 2.7 | 3.1 | 3.0 | 2.8 | 2.4 | 2.5 | 2.5 | 5.8 | 2.2 | 2.9 | 6.7 | 4.6 | 1.2 | 29.2 | 12.3 | 4.3 | 2.3 | 4.3 | 3.7 | 2.22 | 0.67 |
| 32 | 5.9 | 4.1 | 11.1 | 2.7 | 5.0 | 2.3 | 6.0 | 6.0 | 15.4 | 6.1 | 7.5 | 9.6 | 1.1 | 0.8 | 5.4 | 9.3 | 5.7 | 3.6 | 2.3 | 1.13 | 0.36 |
| 33 | 1.4 | 1.3 | 2.0 | 1.1 | 3.1 | 8.3 | 1.1 | 7.0 | 1.2 | 4.1 | 2.8 | 1.3 | 4.2 | 76.0 | 4.6 | 4.6 | 1.5 | 3.6 | 4.0 | 1.07 | 0.46 |
| 34 | 5.8 | 7.7 | 6.1 | 13.8 | 12.3 | 10.6 | 7.8 | 8.6 | 10.8 | 14.9 | 7.0 | 14.5 | 2.7 | 3.5 | 1.0 |  | 11.3 | 15.7 | 1.1 | 1.84 | 0.55 |
| 35 | 11.3 | 10.4 | 9.0 | 14.5 | 7.9 | 15.7 | 11.4 | 12.4 | 13.2 | 11.7 | 6.4 | 11.8 | 3.6 | 19.0 | 17.5 | 9.8 | 4.0 | 12.6 | 3.3 | 3.09 | 0.74 |
| 36 | 12.0 | 9.7 | 8.7 | 13.7 | 10.7 | 11.2 | 11.0 | 10.9 | 17.2 | 11.2 | 11.4 | 12.8 | 1.4 | 51.1 | 14.6 | 10.6 | 3.6 | 11.7 | 3.1 | 2.02 | 0.49 |
| 37 | 6.2 | 6.1 | 6.3 | 6.3 | 5.3 | 8.2 | 5.0 | 7.8 | 6.3 | 6.6 | 6.2 | 5.8 | 3.9 | 50.0 | 19.6 | 4.2 | 3.8 | 8.3 | 3.0 | 3.76 | 0.73 |
| 38 | 14.3 | 11.6 | 13.4 | 13.2 | 11.2 | 16.0 | 15.4 | 13.0 | 18.6 | 13.8 | 11.7 | 13.9 | 4.0 | 13.9 | 8.0 | 12.0 | 6.8 | 12.9 | 2.4 | 3.65 | 0.74 |
| 39 | 7.6 | 14.3 | 14.4 | 18.9 | 8.6 |  | 16.2 | 9.0 | 17.6 | 10.2 | 17.6 | 10.7 | 16.5 | 3.6 | 13.7 | 9.6 | 7.7 | 15.4 | 2.9 | 2.82 | 0.61 |
| 40 | 6.5 | 6.7 | 7.2 | 9.3 | 7.2 | 9.9 | 4.8 | 7.5 | 8.6 | 8.2 | 6.3 | 4.2 | 3.3 | 58.9 | 9.0 | 5.7 | 5.1 | 10.6 | 2.5 | 2.65 | 0.62 |
| 41 | 12.2 | 12.4 | 9.9 | 14.5 | 12.0 | 14.5 | 14.1 |  | 16.0 | 11.5 | 11.3 | 11.5 | 4.8 | 10.3 | 15.7 | 9.8 | 6.0 | 13.3 | 2.8 | 3.79 | 0.79 |
| 42 | 12.1 | 11.4 | 10.9 | 13.4 | 8.6 | 12.3 | 12.3 | 9.5 | 15.1 | 11.3 | 10.6 | 7.8 | 7.7 | 22.4 | 14.9 | 8.8 | 5.0 | 12.5 | 2.8 | 4.55 | 0.85 |
| 43 | 15.8 | 13.8 | 10.0 | 14.3 | 15.0 | 10.9 | 13.3 | 14.0 | 16.3 | 13.5 | 13.4 | 14.9 | 12.6 |  | 1.0 | 12.8 | 7.9 | 17.9 | 1.6 | 2.40 | 0.56 |
| 44 | 3.0 | 2.8 | 4.5 | 2.6 | 3.5 | 3.5 | 2.7 | 5.6 | 1.6 | 3.7 | 7.8 | 2.9 | 1.7 | 11.9 | 1.9 | 4.4 | 2.7 | 3.6 | 3.7 | 2.47 | 0.67 |
| 45 | 6.8 | 11.4 | 11.4 | 13.6 | 8.0 | 10.8 | 11.1 | 7.8 | 14.1 | 8.3 | 13.5 | 5.6 | 9.0 | 15.4 | 12.3 | 9.1 | 4.2 | 10.8 | 3.1 | 3.79 | 0.74 |
| 46 | 11.4 | 13.4 | 13.0 | 15.2 | 10.4 | 10.5 | 15.1 | 8.3 | 15.3 | 12.5 | 15.2 | 8.4 | 11.7 | 3.4 | 9.2 | 13.1 | 8.2 | 14.0 | 2.8 | 3.53 | 0.73 |

**Table 6.** Pearson correlations between diatom indices and physico-chemical parameters of water and sampling station altitudes (* $p < 0.05$, ** $p < 0.01$): Temp.—water temperature, EC—electrolytic conductivity, BOD—biochemical oxygen demand, COD—chemical oxygen demand, TOC—total organic carbon, H'—Shannon Diversity index, E—Evenness index, n—number of cases.

| Diatom Index | Altitude | Temp. | pH | EC | $O_2$ | $BOD_5$ | COD | TOC | Total N | $NH_4^+$ | $NO_2-$ | $NO_3-$ | Total P | $PO_4^{3-}$ | $SO_4^{2-}$ | $Cl^-$ |
|---|---|---|---|---|---|---|---|---|---|---|---|---|---|---|---|---|
| IDAP | −0.32 * | −0.34 * | 0.09 | −0.24 | 0.34 * | −0.57 ** | −0.60 ** | −0.36 * | −0.57 ** | −0.61 ** | 0.01 | −0.18 | −0.32 * | −0.58 ** | −0.13 | −0.30 |
| EPI-D | −0.07 | −0.18 | 0.08 | −0.17 | 0.28 | −0.49 ** | −0.53 ** | −0.41 * | −0.57 ** | −0.55 ** | −0.04 | −0.17 | −0.37 * | −0.56 ** | −0.09 | −0.22 |
| IBD | −0.26 | −0.15 | 0.13 | −0.26 | 0.39 ** | −0.55 ** | −0.56 ** | −0.35 | −0.58 ** | −0.56 ** | 0.10 | −0.19 | −0.40 * | −0.59 ** | −0.13 | −0.25 |
| SHE | −0.02 | −0.23 | 0.03 | −0.20 | 0.23 | −0.45 ** | −0.47 ** | −0.32 | −0.54 ** | −0.56 ** | −0.14 | −0.17 | −0.34 * | −0.55 ** | −0.18 | −0.28 |
| DI-CH | −0.03 | −0.25 | 0.02 | −0.26 | 0.26 | −0.50 ** | −0.53 ** | −0.40 * | −0.52 ** | −0.50 ** | −0.19 | −0.32 | −0.40 * | −0.54 ** | −0.22 | −0.34 |
| IPS | −0.22 | −0.30 * | 0.10 | −0.26 | 0.31 * | −0.53 ** | −0.55 ** | −0.33 | −0.55 ** | −0.58 ** | −0.02 | −0.17 | −0.36 * | −0.56 ** | −0.20 | −0.33 |
| SLA | −0.37 * | −0.29 | 0.02 | −0.14 | 0.35 * | −0.55 ** | −0.59 ** | −0.27 | −0.40 ** | −0.51 ** | 0.02 | −0.11 | −0.29 | −0.53 ** | −0.16 | −0.29 |
| DES | −0.33 * | −0.36 * | 0.08 | −0.36 * | 0.44 ** | −0.65 ** | −0.61 ** | −0.23 | −0.61 ** | −0.62 ** | 0.01 | −0.25 | −0.44 ** | −0.62 ** | −0.31 | −0.44* |
| IDSE | −0.23 | −0.27 | 0.04 | −0.22 | 0.32 * | −0.59 ** | −0.59 ** | −0.25 | −0.56 ** | −0.61 ** | −0.02 | −0.17 | −0.37 * | −0.65 ** | −0.14 | −0.27 |
| IDG | 0.01 | −0.20 | −0.02 | −0.33 * | 0.01 | −0.27 | −0.28 | −0.23 | −0.50 ** | −0.40 * | 0.08 | −0.21 | −0.25 | −0.30 | −0.27 | −0.35 |
| CEE | −0.23 | −0.32 * | −0.09 | −0.27 | 0.26 | −0.46 ** | −0.42 * | −0.14 | −0.55 ** | −0.49 ** | −0.04 | −0.30 | −0.35 * | −0.53 ** | −0.27 | −0.35 |
| TDI | 0.19 | 0.08 | −0.08 | −0.09 | −0.01 | −0.28 | −0.40 * | −0.47 ** | −0.30 | −0.25 | −0.16 | −0.22 | −0.23 | −0.28 | 0.03 | −0.08 |
| %PT | −0.13 | 0.16 | 0.04 | 0.28 | 0.04 | 0.02 | −0.13 | 0.08 | 0.39 * | 0.33 | 0.32 | 0.30 | 0.19 | 0.08 | 0.25 | 0.25 |
| LOBO | −0.07 | 0.06 | 0.11 | 0.14 | 0.09 | 0.08 | −0.08 | −0.21 | −0.20 | 0.02 | −0.03 | −0.01 | 0.09 | −0.21 | 0.26 | 0.25 |
| ROTTt | 0.03 | −0.16 | 0.02 | −0.14 | 0.26 | −0.44 ** | −0.45 * | −0.38 * | −0.45 ** | −0.45 * | −0.15 | −0.17 | −0.39 * | −0.52 ** | −0.06 | −0.14 |
| ROTTs | −0.13 | −0.17 | 0.04 | −0.19 | 0.22 | −0.52 ** | −0.54 ** | −0.33 | −0.57 ** | −0.64 ** | −0.10 | −0.20 | −0.40 * | −0.59 ** | −0.16 | −0.29 |
| WAT | −0.40 * | −0.23 | 0.09 | −0.19 | 0.28 | −0.55 * | −0.58 * | −0.07 | −0.35 * | −0.60 * | 0.22 | 0.01 | −0.18 | −0.48 * | −0.15 | −0.30 |
| TIT | −0.24 | −0.08 | 0.10 | −0.01 | −0.01 | 0.30 | 0.32 | 0.15 | 0.31 * | 0.28 | −0.04 | 0.07 | 0.19 | 0.37 * | −0.13 | −0.03 |
| IDP | −0.15 | −0.37 * | 0.16 | −0.34 * | 0.34 * | −0.52 * | −0.49 * | −0.32 | −0.50 * | −0.57 * | −0.21 | −0.27 | −0.51 * | −0.49 * | −0.41 * | −0.51 * |
| H' | −0.18 | 0.03 | 0.10 | −0.21 | 0.11 | −0.40 ** | −0.35 | 0.01 | −0.46 ** | −0.40 * | 0.30 | −0.15 | −0.15 | −0.32 | −0.12 | −0.24 |
| E | −0.14 | 0.06 | 0.17 | 0.02 | 0.08 | −0.22 | −0.12 | 0.03 | −0.28 | −0.35 | 0.19 | 0.03 | −0.09 | −0.12 | 0.16 | 0.09 |
| n | 45 | 44 | 44 | 44 | 44 | 43 | 31 | 30 | 44 | 28 | 27 | 41 | 40 | 33 | 29 | 29 |

## 4. Discussion

The Sakarya River basin is very diverse in terms of land usage and numerous factors e.g., industrial and domestic discharges, and agricultural runoffs may affect the diatom assemblages. In diatom composition of studied waters species form genera *Navicula*, *Nitzschia*, *Craticula*, and *Amphora* were the most frequent. Most of this species are characterized by occurrence in waters with medium to high trophic level, often saprobity tolerant, up to the β-α level [48]. Oligotrophic and mesotrophic species like for example *Achnanthidium pyrenaicum* [48] were less frequent and occurring mostly in small, undegraded tributaries of the Sakarya River. According to the diatom index values, the lowest values (the worst water quality) were close to big cities (e.g., Ankara) or affected by the presence of extensive agricultural areas in the research area. At these sites, members of the *Craticula* and *Nitzschia* genera and *Fistulifera saprophila*—which are characteristic of strongly polluted, up to polysaprobic, industrial waste waters and heavily degraded environments—were dominated [48]. The water chemistry of the study area show similar regularity of changes as diatom compositions, and was mostly determined by the location of the sampling station in the catchment.

In Turkey, the same analysis of diatom indices and biological water quality assessment were also made in other studies. In a previous study, Solak et al. [31] investigated the source section of river the basin, and the results were similar to the present study: e.g., correlations with the IDAP index and dissolved oxygen or total nitrogen were very similar. However, a comprehensive comparison of the results is impossible due to a small number of analyzed environmental parameters in the cited work. In the present study, most of the evaluated diatom indices were significantly correlated with the important environmental variables at 0.05 and 0.01 importance levels (e.g., biochemical oxygen demand, total nitrogen and phosphate content, phosphates and ammonia ions). Indices such as DES, IDP, and IDAP were correlated with most of the environmental parameters. In contrast, it is reported that the DES index had low correlations with the environmental parameters investigated in the Upper Sakarya Basin [34] and Waal and Vilge Rivers (South Africa) [73].

In a study carried out in the Upper Sakarya Basin [29], *Achnanthidium minutissimum* showed domination, but was not a dominant diatom species in the present study. In earlier classifications this species was considered as indicator of high ecological studies—for IPS index [21] was highly sensitive. According to [48], this taxon is a ubiquitous species, developing in a wide range of environmental conditions. This is why, as pointed out recent studies [12,18], this species should be excluded from water quality assessments in Europe. This issue likely explains why the earlier studies [29] show mainly good and/or high quality status in the Sakarya River. Contrary to previous studies, the present study indicates much lower water quality around the same bodies of water.

The DES and IDAP indices also reported significantly correlated results with the environmental variables at a high confidence level in the rivers of the subtropical zones of Australia [24]. Both IDAP and DES indices are also recommended for coastal zones [74], which are widely represented around Turkey. The third index considered, which has a number of significant correlations, was the Pampean Diatom Index (IDP). This index was developed in Argentina as a specific biotic index for urban, agricultural and industrial impacted waters [71]. The IDP index was correlated with most of the water parameters examined (at *p*-value 0.05). It is important that the IDP index was developed to enable integration of the effect of organic enrichment and eutrophication. The index was improved to distinguish pollution from natural eutrophication phenomena [71]. The Pampean Diatom Index was also successfully applied to other tropical regions. Therefore, it should be noted that the IDP index works both for natural and artificial substratum [75] river basins like the Sakarya, which is anthropogenically transformed and has high naturality. Previous data obtained in Egypt [76] has shown that indices created for similar conditions did not necessarily work in the same specific conditions, while the EPI-D index works very well in Mediterranean rivers [77,78]. One of the most important reasons for limiting the usage of diatom indices adopted from different climate zones (or developed for a specific purpose) is that there are great differences between species structures of diatom assemblages. The important aspect of indices reliability is that the same taxa in various indices have different

sensibilities on pollution or is not included what causes high variability in the obtained scores [12,21]. In our studies an example are values obtained for DES index, which are much higher than other indices. In this case reasons of the differences in indices values were lack of data (for DES index) for such widespread species as *Gomphonema tergestinum* and *Navicula cryptotenella* [21]. In the present study, dominant diatom species occurring in the Sakarya River are also frequent in many European rivers and streams [13,48,79], so important is choosing index comprising ecological data for species from this frequently occurring genera.

The data obtained showed that popular European indices are applicable to Turkish waters. The IPS, IBD, and IDG indices worked with over 90% of identified taxa in the basin, but IDG, as a generic index, is less precise and showed correlation with only three environmental variables.

In European countries, one of the most popular, and important indices is the IPS—Specific Pollution Sensitivity Index [13,14,80–84]. Blanco et al. [85] compared some biotic indices and diatom indices in the Duero Basin (Spain), and they found that the IPS was the best index to reflect the water quality status of the river. Similarly, significant correlations were reported in other studies elsewhere throughout Europe, such as in French rivers [77], Polish springs and rivers [13,14], Belgian and Luxembourg rivers [86,87], Finnish rivers [53], Hungarian rivers [8,88], Estonian rivers [89], and Portuguese [90]. The Water Framework Directive provides a process to ensure the comparability between the biological monitoring results of Member States and their monitoring system classifications. In order to carry out the intercalibration process EU countries organized Geographical Intercalibration Groups, consisting of Member States sharing particular surface water body types. The important point is that IPS is taking into consideration in formula for intercalibration procedure [91]. In both Mediterranean and Continental ecoregions the IPS index is compulsory in water quality assessment for countries like Belgium, Estonia, Luxemburg, Sweden, Bulgaria, Greece, Portugal, Spain, Cyprus [92]. Turkey, as a country trying to join to the European Union have to take into consideration European law. Two countries—Greece and Bulgaria, with similar climate, which are the closest neighbors of Eastern Turkey (our study area) are also use IPS index. In the present study, the IPS index was significantly correlated with eight environmental variables and the can be applied to Turkish waters.

The TIT (Trophic Index of Turkey) was developed specifically for Turkish rivers [55]. In the present work the TIT index was also calculated, and since there were no significant correlations with environmental parameters and the TIT, it is thought that the index did not work in the watercourses of the Sakarya River basin.

## 5. Conclusions

Because Turkey is a very diverse country in terms of natural environments, to obtain reliable results of diatom indices, the test of a few of them is required. The Tropic Index of Turkey was created especially for monitoring water quality in Turkey but from our studies is possible to draw a conclusion that this index does not work properly in this research area. It seems that European diatom indices (e.g., IPS) may be applicable to studied waters in Turkey.

**Supplementary Materials:** The following are available online at http://www.mdpi.com/2073-4441/12/3/703/s1, Table S1: List of sampling stations with abbreviations and geographical coordinates.

**Author Contributions:** C.N.S., Ł.P., É.Á., and G.V. designed the study. H.A.E. and A.M.Y. conducted the field sampling. E.Y. prepared the samples for counting. C.N.S. made qualitative analyses of samples using a light microscope. All authors discussed the results and contributed to the final manuscript. All authors have read and agreed to the published version of the manuscript.

**Funding:** The project, funded by the General Directorate of State Hydraulic Works in the "DSI Capacity Development, and Water Quality Monitoring Project in Sakarya Basin" during 2017 and 2018 and also the Ministry of Science and Higher Education under the name of "Regional Excellence Initiative" in the years 2019–2022 Project No. 026/RID/2018/19.

**Acknowledgments:** The authors wish thank David Duffy for language correction of the manuscript. The authors thank to General Directorate of State Hydraulic Works Investing, Planning and Allocations Department, Environmental Section Managers and Kocaeli University Hydrobiology R&D Laboratory for their valuable support during the sampling and analysis procedure.

**Conflicts of Interest:** The authors declare no conflict of interest.

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
