# Peer review of "Use of Diatoms in Monitoring the Sakarya River Basin, Turkey"

_water, doi:10.3390/w12030703_

Round 1

Reviewer 1 Report

The manuscript “Use of diatoms in monitoring the Sakarya River basin, Turkey” describe the monitoring results of inland waters in a Turkey region comparing several diatom indeces. The overall manuscript is well structured, the text is almost exhaustive, and the statistical analyses are sufficiently descriptive. However, some minor revisions are required.

The English editing seems good; but I feel not qualify to detect the presence of minor mistakes in a text generally good. It seems to me that at:

line 111, “the highest” should be used, instead of “the high” line 287, after TIT a coma would improve the readiness of the sentence

However, I don not have other comments on English editing.

The main misprint in the text is represented by the presentation of the tables. Except for Tables 1 and 2 all the other tables are wrongly cited, numbered and captioned. Please check and correct the sequence, the captions and include the missing table of correlations.

The caption in Fig. 2 should have in italics all the species names.

Considering the content, I would suggest to improve Introduction with a Table resuming which indices were calculated for the present elaboration by Omnidia (the software is widely used but it is not free so not all researchers may be able to use it and to access information), the full names and the scientific references. Moreover, as the WFD is cited I would also include the information on which indices were approved by European Commission to be used to assess river quality. To this purpose, I suggest the authors to consider and cite the COMMISSION DECISION (EU) 2018/229 of 12 February 2018, where for each European geographic ecoregion and for each state the intercalibrated indices are listed. This information will be helpful in better reading the Discussion and can provide more details to understand if the indices in the European Decision are suitable or not for Turkey area. The main example is IPS, that is not only “popular”, but it is compulsory for the assessment of river quality in most of European states, both in Mediterranean and in Continental ecoregions.

I attached a copy of the European decision.

Author Response

First of all, we would like to thank to the reviewers for their valuable contributions.

Line 287, coma was edded after TIT.

Table 3 and 4 was corrected. Also, Table 5 was removed from the text because project leaders and some investigators would like to use the values for further physico-chemical article.

Figure 2 corrected as italic writing of species names.

The list of diatom indices which we used fort he work was added into introduction.

Directive 2000/60/EC and Commision Desicion 2018/229 & 2005/646/EC was used in the text

Reviewer 2 Report

Thank you very much for this work which will support continue monitoring and research in Turkey, specifically regarding water quality of rivers, streams and tributaries. This work will definitely contribute to this aim an support future work that will be undertaken. 

However, I found there quite a few things that should be improved before the paper is published. For example, the brevity of the introduction takes away the scientific soundness presented in the methods and results sections.

An important issue is that many abbreviations of the indices were not undisclosed, so the reader is left in limbo trying to figure out or searching elsewhere what index is the one presented in the paper. Titles of Tables and Figures were not always clear. For example, Table 3.  is entitled "Diatom taxa recorded during studies" but all the data in this table are water quality data! and Table 4. talks about diatom indexes with some colour coded and all it shows is a list of species in three columns. This is a major error and leads to confusion in understanding what the authors are addressing or trying to say.

References need updating, there are only two references from 2017, one from 2016, two from 2014...most are from early noughties or previous decade. Needs further reading of current literature on this topic. The analysis, interpretation and presentation of results can be improved. Many statistical analysis were carried out however, no values of the eigen values are shown.

Author Response

First of all, we would like to thank to reviewers for their valuable contributions.

The introduction was improved by adding some EU Comission reports on the topic. For material methods, the table which used to calculate diatom indices values was added into the text.

The tables were replaced and corrected following to the text and two tables was given as supplementary materials. The abbreviations of diatom indices were given in Table 1 (List of diatom indices calculated for present work).

References revised and about 20 more current literatures and two EU Comission desicions were used in the text. By this way, the presentation of results was tried to be improved.

Reviewer 3 Report

Dear authors,

Please find below some comments that I wish will help improve your manuscript.

The introduction is too short and lacks structure. It is just a series of sentences with keywords without any logic coherence. You tried to provide some information on previous studies performed in Turkey but that is quite superficial. A complete rewriting of the introduction is required.

The methods section needs to include more information, especially regarding the calculation of the diatom indices, which are a key component of tour study.

Your results section is drafty, and some data is missing (correlations). You call the wrong tables, which is misleading.

The discussion section is also patchy. It includes several ideas in a row but without a real in-depth analysis. You provide several references to other studies but the connection with the present study is hard to find.

The conclusion is supposed to summarize and give the home message from the findings of the study. Under its current form, it reads more like a final comment than a conclusion.

Other comments:

Table 1 does not provide much insightful information. It can be moved to supplements.

L. 73: which studies?

L.79: readers may want to get a deeper understanding of the diatom indexes you used. What re the particularities of each of them? Why using different ones? How do you calculate them? Etc.

L.80: requires more explanation.

L.84: below surface film?

L.84: how were samples collected?

L. 88-93: don´t mix DCA info with CCA info.

L. 89: why do you need to run correlations at both confidence levels?

L. 97: why not testing for correlations amongst variables?

Are you sure Table 3 is properly entitled?

Figure 2: put all scientific names in italics.

L. 116-118: why excluding these two species?

L.118: under which criteria?

L.139-140: unclear.

L. 143-144: Unclear. Which results support your claims?

You may want to spend a few more lines on the CCA results. Maybe include the figure as supplement.

Figure 3: what are the green dots?

L.169-171: contradictory information.

L.172: you say Table 4 includes indices, but I see a list of species. You meant table 5, I guess.

Table 5 legend says it includes Pearson correlations but I see values of diatom indexes. Where are the correlations?

L.175: what do you mean by “the indices worked”?

Table 5 shows a large variability amongst indices for each station. You may want to discuss this.

L.194-200: I don´t see the point in comparing indices.

L.233-235: Confusing. You start talking about the water chemistry but them you jump into the index values.

L.240: “Worked the upper part”?

L.243-249: cannot trust these statements as the results are missing.

L.250-252: so?

Page 12 is dense, with lots of references to other studies but the conductive thread is hard to follow.

Author Response

First of all, we would like to than kto reviewers for their valuable contributions.

The introduction and methods chapter were redrafted and same requred information were added.

Missing correlations table was added . Table and Figures capion have been corrected. Discussion chapter was redrafted and expanded.

The conclusions were shortened and clarified.

Table 1 was moved to supplementary materials.

73: We added this information 79. We added same information to methods and also addition table with list of used indices was included.

L.80. We include same additional explanation to methods.

L.84. Our mistake + this information was deleted.

88-93: We put this information in separated paraghraph. 89: Two confidence levels were used for test which factors are the most importatnt for indices interpretaion. In biological/environmental studies is standard now.

L.97: The test was perform using CCA analysis but the results were not statistically significat so we consedered not includ it to the paper.

Tables and Figures capion have been corrected

Figure 2: The italics text were use for latin name.

116-118: We added information why this two species were excluded from analysis.

L.118. We added this information to methods.

139-140. We clarified this sentence. 143-144. Base on location stations in DCA graph and chemistry table. The CCA analysis was not statistically significat so we consedered not includ it to the paper.

Figure 3: We added description to this figure.

169-171. We clarified this sentence.

L.172:, Table 5, Tables and Figures capion have been corrected

175. We clarified this sentence and adeed same additiona comment to discussion about indices variabilit.

L.194-200. Ths is general information about obtained results and ecological status of water studed.

L.233-235, L 420,  L.243-249, L.250-252: ,  Discussion chapter was redrafted and expanded

Round 2

Reviewer 2 Report

Thank you for carrying out many of the suggested reviews and for including more recent references.

However, the abstract still have undisclosed abbreviations. 

The introduction is still too weak, poor, to be considered for publication for a scholarly scientific article and should be improved further. The authors have include some new references, for example, references from the European Framewater Directive but this does not provide sufficient basis for a scholarly written introduction. 

The syntaxis of the English language should be improved  for the introduction and for the whole paper. It is very difficult to follow in some parts, introduction and presentation of the results.

Table 3. there are not units for the physical and chemical parameters included, please include them.

Author Response

We would like to thank for your valuable comments. We tried to improve our manuscript following your suggestions. The text rechecked also by a native speaker. Please kindly accept my apologizes for our simple linguistic mistakes. 

Detailed list of changes is included in the paper document (tracking changes option) and listed below.

  1. Introduction, abstract and language were  improved according Reviewer suggestions.
  2. Figure 1 was replace – one sampling station had double signature.
  3. Captions of table 3 and 6 were updated. Missing informations were added.

References list was improve and updated.
